# Migration and interbirth transition rate using Benin Demographic and Health Survey data: Does episode-splitting matter?

**Boladé Hamed Banougnin**[1][☯]*, **Oluwaseyi Dolapo Somefun**[2][☯]*, **Abibatou Agbéké Olakunle**[3][☯]

**1** Centre for Social Science Research, University of Cape Town, Cape Town, South Africa, **2** Women´s Health Research Unit, University of Cape Town, Cape Town, South Africa, **3** École Nationale Supérieure de Statistique et d'Économie Appliquée, Abidjan, Côte d'Ivoire

☯ These authors contributed equally to this work.
* seyi.somefun@gmail.com (ODS); bolade.banougnin@uct.ac.za (BHB)

## Abstract

### Background

The relationship between migration and fertility has vexed demographers for years. One issue missing in the literature is the lack of careful temporal consideration of when women migrate and specifically, the extent to which they do either before or after live births.

### Objective

Here, we opt for a more appropriate methodological approach to help remedy the complexity of the temporal aspect of migration and childbirth processes: regression models using the episode-splitting method.

### Methods

This paper applies a rarely used methodological approach (episode-splitting) in the literature of migration-fertility relationship to investigate how internal in-migration is associated with inter-birth intervals among women in Cotonou, the largest city of Benin. Data comes from the 2017–2018 Benin Demographic and Health Survey (DHS) of women aged 15–49. Estimates from exponential regression models with episode-splitting were compared to estimates from exponential regression models without episode-splitting approach. Sensitivity analysis was also conducted to determine the robustness of the comparison between the two methods. Akaike Information Criteria (AIC) and Bayesian Information Criteria (BIC) were used to identify the method that provides models with best fit.

### Results

The results from (standard) exponential regression models without episode-splitting show that there is no significant association between migration and interbirth transition rate. However, significant associations between migration and interbirth transition rate emerge after applying the episode splitting method. The hazard ratios (HR) of the transition to the next

**Data Availability Statement:** Data is publicly available on the Demographic and Health Survey website (www.measureDHS.com).

**Funding:** The author(s) received no specific funding for this work.

**Competing interests:** The authors have declared that no competing interests exist.

live birth are higher among migrant women than among nonmigrant women. This trend is persistent even after 10 years spent in Cotonou by migrant women.

## Conclusion

Exponential regression models with episode-splitting were of better fit than exponential regression models without episode-splitting. Sensitivity analysis conducted seems to confirm that models with episode-splitting produce estimates that are accurate, reliable and superior to models without episode-splitting. The results suggest a long-run process adaptation of migrants to lower fertility behaviours in Cotonou and are therefore consistent with the socialization hypothesis.

## Introduction

Several studies have emphasized the heterogeneity in the fertility of migrant women [1]. While there has been huge focus on international migration and fertility, the high fertility rates and rapid urbanization of sub-Saharan Africa (SSA) have important implications for health and development in the region—raising questions of how domestic migration, particularly rural-to-urban migration, informs fertility-related behaviours. Overpopulation has been linked with increasing urban slums and other unpleasant conditions in cities, which generally increases the disease burden in urban centres [2]. In addition, there is a pressure on little resources available in the urban areas. These are interrelated with fertility and future fertility trends. The total fertility rate in Benin was 5.7 children per woman in 2018 (including 4.0 in Cotonou, 5.2 in other urban areas, and 6.1 in rural areas), which is higher than the SSA region average [3].

A number of studies [4–6] have examined the determinants of fertility and several interventions aimed at reducing the number of births per woman have been put in place [7, 8]. The role of internal migration on fertility has also been studied and the results have been mixed [9, 10], leaving questions of the effect of migration on fertility-related behaviours. Notably, the issue of temporal precedence between migration and fertility has been far less investigated, and even less so with cross-sectional data, which continues to muddy understanding of the linkages.

The use of time-dependent covariates in transition rate models is almost inevitable in event history analysis for all disciplines. Explicit measurement and inclusion of time-dependent covariates in such models represent an important step for the empirical study of social change. Demographers have frequently used time-dependant covariates like migration, marriage, and wealth to explain fertility [11–14]. However, they used methods of analysis (which include event history analysis methods) that do not carefully consider the time-dependent nature of such covariates. Some have partially cancelled out the time-dependent nature of these covariates by focusing solely on the period after the event of interest occurred [11, 13]. In doing so, their analyses do not adequately capture the effects of change in one phenomenon (e.g., migration, marriage, and wealth) over time on change in another phenomenon (e.g., fertility) over time. Based on this limitation, researchers have recommended the application of time-dependent covariates in the framework of parallel processes [15, 16].

The parallel processes can operate either at the (1) individual level (e.g., how women's migration influences their fertility), (2) intermediate level (e.g., how changing household structure affects women's labour force), (3) macro level (e.g., how a new population policy influences fertility decline), or (4) any combination of the aforementioned levels (e.g., analyses of

life course, cohort, and period effects supposed to simultaneously include time-dependent covariates at different levels). The main methodological issue of either of these systems of parallel processes is the reverse causation [17]. Reverse causation refers to a situation in which the dependent process precedes and causes the independent covariate process instead of the other way around [12, 18, 19].

For example, the analysis of the influence of migration on fertility as studied by demographers clearly leaves open the potential for reverse causation [9, 20–26]. In migration-fertility relationship studies, the internal time-dependent covariate ($X_t$) and the fertility over time ($Y_t$) describe stochastic processes. When fitting a causal model, the stochastic process $X_t$ affects $Y_t$, which in turn, affects $X_t$. The effects operate either directly ($X_t$ affects $Y_t$ and $Y_t$ affects $X_t$) or indirectly with "feedback" effects ($Y_t$ affects $Y_t$ via $X_t$, and $X_t$ affects $X_t$ via $Y_t$). For instance, researchers suggest that childbirths are influenced by migration history and that having large family size can lead to migration—especially selective migration from a lower-fertility setting to a higher fertility setting [27]. These interdependence processes [28] are what Tuma and Hannan (1984) called a "dynamic system". The methodological literature proposes at least two major approaches to study dynamic systems, mainly the dynamic system approach and another approach that takes into consideration temporal precedence and closer to causal approach [29, 30].

This study opted for the approach that considers temporal precedence, especially the episode-splitting method to examine how migration to Cotonou (the largest city of Benin) is associated with progression to next birth. In other words, the current study applies the episode-splitting method to investigate how internal in-migration are associated with interbirth transition rate among women in Cotonou [17]. This paper acknowledges the data limitations of previous studies by using a different method to examine these associations. It contributes to the literature related to issue of temporal precedence when analysing the association between migration and fertility.

Like many other sub-Saharan African countries, Benin Republic has a fast-growing population. Reports from the last two population and housing censuses show that its population increased from 6.7 million in 2002 to 10 million in 2013; that is a population growth rate of 3.5% per annum [31]. This population growth rate is above the sub-Saharan Africa average population growth rate—which was 2.5% per annum during the period between 2005 and 2010 [32]. The paper focuses specifically on Cotonou which is the largest city and the economic hub of Benin Republic. The population of Cotonou grew from about 20,000 inhabitants in 1950 (United Nations, 2018) to 679,012 inhabitants in 2013 [31]—and will reach about 1,055,000 inhabitants by 2035 according to demographic projections of the United Nations (2018) [33]. This suggests that Cotonou is under excessive pressure in terms of space occupation. Cotonou is also the most densely populated city in Benin Republic with 8,595 inhabitants per square kilometre according to the (last) 2013 National Population and Housing Census [31].

Episode splitting method is more adequate to estimate the effects of internal time-dependent processes on transition rates [34]. Applying episode-splitting method to identify the effect of migration on fertility can be described as follows: Migration status (here the internal time-dependent covariate) changes its value only at discrete point in time. At all points in time when migration status changes its value (e.g., from the status of non-migrant (= 0) to the status of migrant (= 1)), the original childbirth episodes are split into sub-episodes. Within the new sub-episodes, the date of migration lies between the starting and ending times of a birth occurrence. Fig 1 (below) shows a representation of how a change in migration status could affect the occurrence of additional births over time.

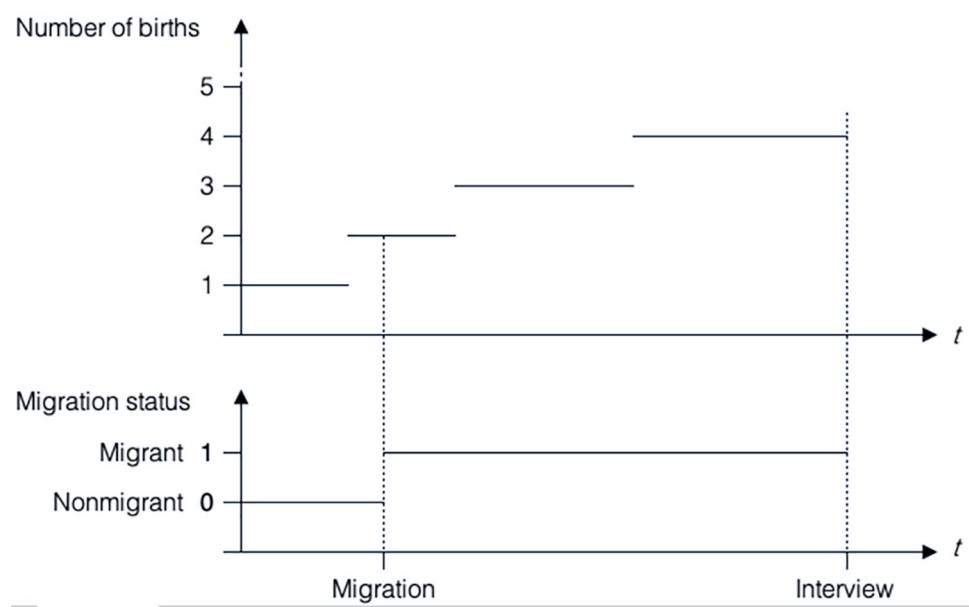

**Fig 1. Effect of change in migration status on the occurrence of additional births over time.**

## Data and methods

### Data

This study uses data from the 2017–2018 Benin Demographic and Health Survey (DHS) for the implementation of the model fit. The 2017–2018 Benin DHS used a two-stage cluster sample design with Cotonou, urban areas outside of Cotonou, and rural areas as strata. The DHS selected 555 enumeration areas (EAs) based on the most recent (2013) Benin census frame, and randomly selected 26 households within each EA. Fifty-two EAs were selected in Cotonou. The DHS Program made available many dataset types including PR (Household members Recode) and BR (Births Recodes) that are used in the current study. The BR dataset contains (among others) the full birth history of all women aged 15–49 years while the PR includes characteristics of all household members. In contrast to many DHS datasets, the 2017–2018 Benin DHS collected information on lifetime migration for all household members. Of all variables on lifetime migration, those on the date of migration and origin places are sufficient for the fit of episode-splitting models. Identification variables (cluster number, household number, and household member's line number) enable to merge the lifetime migration variables (from the PR dataset) to those related to women's birth episodes (from the BR dataset). The unit of analysis is live birth to woman interviewed in Cotonou whether in-migrant or non-migrant.

The initial sample was 2331 live births to 779 women living in Cotonou. Of the 779 women, we dropped 51 live births to 16 women who were living somewhere else but spent less than six months in Cotonou—to be consistent with the definition we provide later—yielding 2280 live births to 763 women. We then excluded 97 multiple live births yielding a final sample of 2183 live births to 753 women.

### Measures

The dependent variable of the analysis is the interbirth interval defined as the time between two successive live births to a woman. In DHS data, interbirth interval are expressed in

months. The main independent variable is the migration status. Here, migrants are women who moved from any other place (either other urban areas or rural areas) in Benin to Cotonou at least six months preceding the interview. This was based on the definition adopted during the 2018 Benin Demographic and Health Survey data collection. The threshold of six months is the one adopted by the Benin national institute for statistics and economic studies during the 2018 DHS (INSAE & ICF, 2019). Control variables include: the occurrence of any previous child death, woman's age, household wealth quintile, and educational level.

## Methods and specific models

**Methods.** The transition rate model used with the episode splitting method is the exponential model for its simplicity. Let $t_{ij}$ be the birth interval (in months) of the $j^{th}$ birth to the $i^{th}$ woman, and $T$ its corresponding exponential distribution with $\lambda$ constant hazard. The density function $f(t)$, survivor function $G(t)$, and transition rate—from the $j^{th}$ birth to the $(j+1)^{th}$— model $r(t)$ of $T$ are respectively

$$f(t) = a\exp(-\lambda t) \quad \lambda > 0 \tag{1}$$

$$G(t) = \exp(-\lambda t) \tag{2}$$

$$r(t) = \lambda \tag{3}$$

Then, consider one of the original episodes of the 4-uplet $(u, v, s, t)$ with origin and destination states $u$ and $v$ with starting and ending times $s$ and $t$, respectively. It is assumed that the specific episode is defined on a process time axis so that $s = 0$. Now assume that this episode is split into $L$ sub-episodes

$$(u, v, s, t) \equiv \{(u_l, v_l, s_l, t_l)|l = 1, \ldots, L\} \tag{4}$$

The likelihood of this episode can be written as the product of the transition rate $r(t) = \lambda$ (3), and the survivor function $G(t)$ (2). Only $G(t)$ is affected by the process of episode splitting and can be written as a product of the conditional survivor functions for each split of the episode:

$$G(t) = \prod_{l=1}^{L} G(t_l|s_l) \tag{5}$$

with conditional survivor functions defined by

$$G(t_l|s_l) = \exp\left\{ -\int_{s_l}^{t_l} r(\tau)d\tau \right\} \tag{6}$$

On the right-hand side, the transition rate $r(\tau)$ could be for each split $(u_l, v_l, s_l, t_l)$ and may depend on covariate values for this split. The general likelihood function for transition rate models combined with conditional pseudosurvivor functions, assuming a given origin state, is

$$\mathcal{L} = \prod_{v \in \mathfrak{D}} \prod_{j \in \varepsilon_v} r_v(t_j) \prod_{j \in \mathcal{N}} \tilde{G}_v(t_j|s_j) \tag{7}$$

Therefore, it is possible to use this likelihood function with a sample of original (not split) episodes in which all starting times are zero, instead of a sample of splits (Blossfelt et al., 2019). Although the episode-splitting approach is not new in the framework of survival analysis methods, it is almost not used (to the best of our knowledge) in the literature of migration-

fertility relationship. A recent paper on how internal migration affect short and long interbirth intervals did not use the episode-splitting approach on the competing-risk modelling while data permitted to split migration episodes [10]. Due to the 2017–2018 DHS sample design (which is a two-stage probability sample design), the study applies simple weighting for estimates of proportions. Complex sample design is applied for computations requiring standard errors, confidence intervals or significance testing. Analysis from this paper is performed with StataMP 16.

**Models.**   The study used exponential models both on the original data (without episode-splitting) and data after episode-splitting. It compares estimates from exponential regression models using data without episode-splitting (i.e., ignoring the time-varying nature of migration) and estimates from exponential regression models using data with episode-splitting (i.e., accounting for the time-varying nature of migration). The idea of the episode splitting method adopted in this study can be described as follows: migration status changes its value only at discrete points in time. At all points in time, when migration status changes its value, the original migration episode is split into pieces—called splits (of a migration episode) or sub-episodes. For each sub-episode a new record is created containing (1) information about the origin state of the original migration episode, (2) the value of the migration status at the beginning of the sub-episode, (3) the starting and ending times of the sub-episode (information about the duration would only be sufficient in the case of an exponential model), and (4) Information indicating whether the sub-episode ends with the destination state of the original episode or is censored. All sub-episodes, apart from the last one, are regarded as right censored. Only the last sub-episode is given the same destination state as the original episode. Exponential models are fitted for the analysis.

The multivariable analysis is based on four separate models. The first model analyses the association between migration and interbirth transition rate and the second model adjusts for control variables (the occurrence of any previous child death, woman's age, educational level, and household wealth quintile). The third model is similar to the first model except for the fact that the migration variable (from the first model) is replaced by a more detailed migration variable with duration of migration (for migrants and return migrants). The last model is equivalent to the third model but includes the control variables. The introduction of control variables aims at asking the question of how the effect of migration (including migration by duration) changes after other relevant covariates are added in the crude model. These models are fitted considering interbirth interval by ever-migrant status (i.e., migration variable without episode-splitting) on the one hand, and interbirth interval by migrant timing (i.e., migration variable obtained after episode-splitting) on the other hand.

**Sensitivity analysis model.**   The DHS program has many processes in place to aid women's recall months and years of demographic events. For example, enumerators are asked to record months and years of live births asking from the most recent live birth to the less recent one. But there are still concerns that women do not accurately recall the exact months and years of their live births. The same concerns hold for the date of migration, especially if another household member answers on the woman's behalf. Recalls of the months and years of births and migration are of major concerns for the validity of our comparative analyses. We therefore conducted a sensitivity analysis to see if the analyses above changed when we limit the sample of the study to women who are more likely to reduce the recall biases. A recent study carried out by Larsen et al. (2019) found that recall biases are high among poorer and less-educated populations [35]. In this study, we fit the above models excluding women with no education. (Findings from these sensitivity analyses are shown in Table 4). In the sensitivity analysis, we checked whether our conclusions from the comparison of performance between models with and without episode-splitting differed from the main results.

## Results

Table 1 presents the sample description of the study. Women included in this study were 33.4 years old on average with a standard deviation of 7.9 years old. More than three quarters of them (575 out of 753) had at least two live births at the time of the survey. For those women, the average interbirth interval is of 43.3 months with a standard deviation of 28.8 months. More than 59% of the women living in Cotonou are ever migrants. Only 9% of women living in Cotonou at the time of the survey had a higher level of education. The remaining 91% included 30% with no education, 30% with a primary level of education, and another 31% with a secondary level of education.

Fig 2 presents means and 95% confidence intervals of interbirth intervals according to ever-migrant status (from data before episode-splitting) and migrant timing (from data obtained after episode-splitting) (see Table 2 for more details). Part of Table 2 is plotted on Fig 2. Fig 2A shows that births to women who have never migrated and those who have ever-migrated in Cotonou are approximatively of the same mean interval length (about 42 months). Fig 2B shows that the mean birth interval is higher among migrants (42.2 months) than among non-migrants (39.6 months). Among births to migrant women, the mean interval is nearly 8 months longer for births that occur after migration (42.8 months) than for births that occur before migration (34.6 months).

Table 3 presents results from exponential models for data before episode-splitting and data after episode-splitting. AIC and BIC are also presented for evaluations basis. (AIC and BIC statistics suggest that models from episode-splitting are of better fit.) Findings from data before episode-splitting suggest that there is no significant association between having ever migrated and interbirth transition rate. In contrast, after episode-splitting, the effects of migration on interbirth transition rate became clearly and consistently significant. The interbirth transition rate increased by 17% (Table 3, Model 1A) for migrant women, and by 16% (Table 3, Model 1B) after controlling for education, household wealth quintile, previous child death, and age. The interbirth transition rate decreases with the time spent by migrants in Cotonou.

Compared to non-migrant women, the hazard ratios of the transition to the next live birth are higher among migrant women who spent less than five years in Cotonou (Model 2A: HR = 2.62, 95% CI 2.15–3.19; Model 2B: HR = 2.38, 95% CI 1.92–2.94), migrant women who spent between five and ten years in Cotonou (Model 2A: HR = 1.38, 95% CI 1.22–1.56; Model 2B: HR = 1.30, 95% CI 1.15–1.48), and migrant women who spent more than ten years in Cotonou (Model 2A: HR = 1.09, 95% CI 1.01–1.19, Model 2B: HR = 1.10, 95% CI 1.02–1.19). Furthermore, based on AIC and BIC, models from episode-splitting data are of best fit than models from data before episode-splitting.

**Table 1. Sample description of the study.**

|  | N/ Mean (%/ SD)—N = 753 | Missing/NA |
|---|---|---|
| Interbirth interval | 43.3 (28.8) | 178 |
| Migration status |  | 0 |
| Non-migrant | 307 (40.8%) |  |
| migrant | 446 (59.2%) |  |
| Highest educational level |  | 0 |
| no education | 227 (30.1%) |  |
| primary | 228 (30.3%) |  |
| secondary | 232 (30.8%) |  |
| Higher | 66 (8.8%) |  |
| Age | 33.4 (7.9) | 0 |

NA: not applicable.

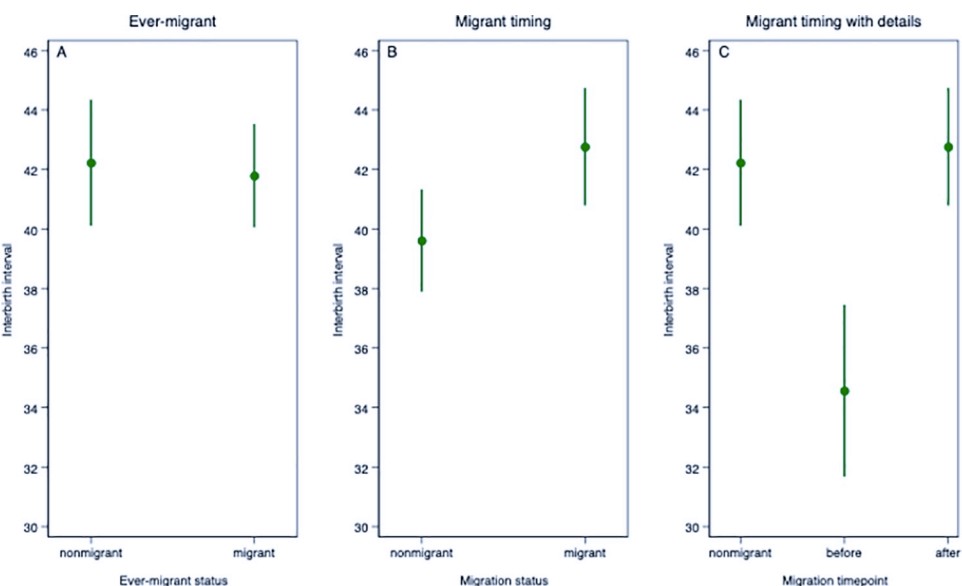

**Fig 2. Means and 95% confidence intervals of interbirth intervals by ever-migrant status and migrant timing.**

**Table 2. Mean (standard deviation) of interbirth intervals, by selected characteristics of live births, according to migration status, before episode-splitting and after episode-splitting.**

| Variable | Before episode splitting | | | | After episode-splitting | | | | | |
|---|---|---|---|---|---|---|---|---|---|---|
| | N | Migration status | | Total | N | Live birth to women | | (3) = (1) & (2) | (4) after migration | Total |
| | | Nonmigrant | Migrant | | | (1) never migrant | (2) before migration | | | |
| Highest educational level | | | | | | | | | | |
| no education | 548 | 41.4 (2.2) | 41.3 (1.4) | 41.3 (1.2) | 594 | 41.4 (2.2) | 35.4 (2.0) | 37.9 (1.5) | 43.2 (1.7) | 40.6 (1.1) |
| primary | 490 | 40.9 (1.6) | 41.6 (1.8) | 41.2 (1.2) | 520 | 40.9 (1.6) | 32.7 (3.4) | 38.8 (1.5) | 42.5 (1.9) | 40.2 (1.2) |
| secondary | 316 | 43.8 (2.3) | 43.7 (2.6) | 43.7 (1.7) | 339 | 43.8 (2.3) | 37.9 (3.9) | 42.5 (2.0) | 43.2 (2.9) | 42.8 (1.7) |
| higher | 71 | 47.0 (5.7) | 40.2 (3.8) | 43.0 (3.3) | 78 | 47.0 (5.7) | 23.5 (6.4) | 40.5 (4.9) | 39.6 (4.1) | 40.0 (3.2) |
| Experienced child death | | | | | | | | | | |
| no | 1334 | 42.8 (1.2) | 42.3 (1.0) | 42.5 (0.8) | 1437 | 42.8 (1.2) | 35.1 (1.7) | 40.2 (1.0) | 43.0 (1.2) | 41.4 (0.8) |
| yes | 91 | 34.6 (2.7) | 35.4 (2.3) | 35.1 (1.8) | 94 | 34.6 (2.7) | 27.9 (4.0) | 32.1 (2.3) | 38.9 (2.3) | 34.7 (1.7) |
| Age group | | | | | | | | | | |
| 15–24 years | 43 | 24.3 (1.3) | 29.8 (2.2) | 27.4 (1.5) | 47 | 24.3 (1.3) | 21.2 (4.7) | 23.4 (1.7) | 30.1 (2.2) | 26.2 (1.5) |
| 25–39 years | 855 | 40.8 (1.3) | 40.0 (1.2) | 40.3 (0.9) | 921 | 40.8 (1.3) | 32.7 (2.0) | 38.4 (1.1) | 40.5 (1.3) | 39.4 (0.9) |
| 40–49 years | 527 | 46.1 (2.2) | 45.6 (1.8) | 45.8 (1.4) | 563 | 46.1 (2.2) | 37.2 (2.5) | 42.5 (1.7) | 48.4 (2.2) | 44.8 (1.3) |
| Total | 1425 | 42.2 (1.1) | 41.8 (1.0) | 42.0 (0.8) | 1531 | 42.2 (1.1) | 34.6 (1.6) | 39.6 (0.9) | 42.8 (1.1) | 41.0 (0.7) |

N = number of live births to women aged 15–49 years; values in parentheses are standard errors of means.

**Table 3. Exponential models presenting hazard ratios and 95% Confident Intervals (CIs) before and after episode-splitting for the relationship between migration and fertility.**

| | Before episode-splitting–ever migrant | | | | After episode-splitting–migration timing | | | |
|---|---|---|---|---|---|---|---|---|
| | Model 1a | Model 1b[†] | Model 2a[†] | Model 2b | Model 1A | Model 1B[†] | Model 2A[†] | Model 2B |
| Migration [a] | | | | | | | | |
| Nonmigrant (Ref.) | 1 | 1 | | | 1 | 1 | | |
| Migrant | 1.01 (0.94–1.08) | 1.00 (0.93–1.07) | | | 1.17*** (1.09–1.27) | 1.16*** (1.08–1.25) | | |
| Migration [a] (with duration) | | | | | | | | |
| Nonmigrant (Ref.) | | | 1 | 1 | | | 1 | 1 |
| Migrant | | | | | | | | |
| <5 years | | | 0.98 (0.84–1.14) | 0.94 (0.81–1.09) | | | 2.62*** (2.15–3.19) | 2.38*** (1.92–2.94) |
| 5–10 years | | | 1.07 (0.96–1.20) | 1.01 (0.91–1.13) | | | 1.38*** (1.22–1.56) | 1.30*** (1.15–1.48) |
| >10 years | | | 0.99 (0.92–1.07) | 1.01 (0.93–1.09) | | | 1.09** (1.01–1.19) | 1.10** (1.02–1.19) |
| AIC | 2261.37 | 2263.62 | 2264.63 | 2267.36 | 2255.21 | 2258.18 | 2246.57 | 2253.62 |
| BIC | 2271.89 | 2316.24 | 2285.68 | 2330.51 | 2265.87 | 2311.52 | 2267.91 | 2317.62 |
| Number of women (weighted) | 988 | 988 | 988 | 988 | 988 | 988 | 988 | 988 |

[a] The migration variable is meant for ever migrant when using data before episode-splitting whereas it is meant for migration timing when using data after episode-splitting

† model controlled for education, household wealth quintile, age, and previous child death experience of the mother

***$p < 0.01$

**$p < 0.05$

*$p < 0.1$; Confidence intervals between parentheses; AIC = Akaike information criterion; BIC = Bayesian information criterion; Ref.: reference category.

## Sensitivity analysis

When excluding women with no education from the sample, (Table 4, N = 644), we observed no statistically significant differences in the relationship between migration and interbirth transition rates as compared with our main analysis. We found only one case in which association that was not significant in the main analysis became slightly significant in the sensitivity analysis. Indeed, there is a slight difference between Model 2a (Table 3) of the main analysis and Model 2a (Table 4) of the sensitivity analysis. Being ever migrant (taking into account the duration) was not significantly associated with interbirth transition rate in the main model (Model 2a in Table 3). However, the sensitivity analysis model (Model 2a, Table 4) showed that the hazard ratios of interbirth transition rate is statistically significantly higher among ever migrant women who spent 5–10 years in Cotonou (HR = 1.13, 95% CI 0.99–1.28) than among women who have never migrated. Results between the main analysis and the sensitivity analysis were largely consistent. As for the main analysis, AIC and BIC statistics suggest that models from episode-splitting are of better fit.

## Discussion

This present study examined the association between internal migration in Cotonou (Benin) and birth intervals. We compared exponential models using episode-splitting method to standard exponential models estimates. Standard exponential models were fit on data before episode-splitting from live births to women of reproductive age. Exponential models using

**Table 4. Results from sensitivity analysis—exponential models presenting hazard ratios and 95% Confident Intervals (CIs) before and after episode-splitting for the relationship between migration and fertility among women with primary level of education or higher.**

| | Before episode-splitting–ever migrant | | | | After episode-splitting–migration timing | | | |
|---|---|---|---|---|---|---|---|---|
| | Model 1a | Model 1b† | Model 2a† | Model 2b | Model 1A | Model 1B† | Model 2A† | Model 2B |
| Migration [a] | | | | | | | | |
| Nonmigrant (Ref.) | 1 | 1 | | | 1 | 1 | | |
| Migrant | 1.01 (0.92–1.10) | 1.00 (0.92–1.09) | | | 1.16*** (1.06–1.28) | 1.14*** (1.04–1.25) | | |
| Migration a (with duration) | | | | | | | | |
| Nonmigrant (Ref.) | | | 1 | 1 | | | 1 | 1 |
| Migrant | | | | | | | | |
| <5 years | | | 0.96 (0.75–1.21) | 0.95 (0.76–1.18) | | | 2.19*** (1.78–2.69) | 2.23*** (1.81–2.74) |
| 5–10 years | | | 1.13* (0.99–1.28) | 1.06 (0.93–1.20) | | | 1.27*** (1.11–1.46) | 1.28*** (1.11–1.47) |
| >10 years | | | 0.97 (0.88–1.07) | 0.98 (0.89–1.08) | | | 1.08 (0.97–1.19) | 1.08** (0.97–1.19) |
| AIC | 1479.77 | 1483.35 | 1482.29 | 1486.96 | 1476.21 | 1480.61 | 1472.96 | 1479.92 |
| BIC | 1489.32 | 1526.34 | 1501.40 | 1539.50 | 1485.89 | 1524.19 | 1492.33 | 1533.19 |
| Number of women (weighted) | 644 | 644 | 644 | 644 | 644 | 644 | 644 | 644 |

[a] The migration variable is meant for ever migrant when using data before episode-splitting whereas it is meant for migration timing when using data after episode-splitting

† model controlled for household wealth quintile, age, and previous child death experience of the mother

***$p < 0.01$

**$p < 0.05$

*$p < 0.1$; Confidence intervals between parentheses; AIC = Akaike information criterion; BIC = Bayesian information criterion; Ref.: reference category.

episode-splitting models were fit on data from births to women of reproductive age with additional observations depending on whether births occurred after migration to Cotonou. We have also conducted a sensitivity analysis that seems to confirm that the episode-splitting method produces estimates that are accurate, reliable and superior to the standard exponential method.

The results from standard exponential models show that there is no significant association between migration and interbirth transition rate. However, significant associations between migration and interbirth transition rate emerge after applying the episode splitting method. Findings from exponential models with episode-splitting method reveal that migrant women had a higher interbirth transition rate than non-migrant women in Cotonou. We have also used AIC and BIC estimates from models with and without episode-splitting to identify models from which method provide better fit. We found that estimates from models using episode-splitting method were of better fit than models without episode-splitting. Consequently, the results clearly demonstrate the need to adopt this modelling strategy to appropriately control for internal time-dependent process. In our analysis, the variable on the duration of migration in Cotonou was used to split migration episodes. Findings from episode-splitting method are similar to other studies [27] which took into account temporal precedence in analyses using time-dependent covariate. Previous studies on the relationship between migration and fertility have reduced bias due to temporal precedence by considering only births that occurred at the destination place [10, 11]. In the absence of purely longitudinal data, the technique of episode-splitting is a superior alternative to accounting for temporal precedence.

The models using episode-splitting methods was fit to determine the relationship between migration and fertility in Cotonou. We found that migrant women had higher hazard ratios of transition to a next birth than non-migrant women—and even after controlling for age, education, and child death. Then, the higher the time spent by migrant women in Cotonou, the lower the transition rate to a next birth. Our results confirmed the gradual adaptation and socialization hypothesis stated in previous studies carried out in Cotonou, Benin [36]. Migrants from high-fertility settings to low-fertility settings are more likely to maintain their high-fertility behaviours and adjust to lower-fertility behaviours with time passing.

Furthermore, in this study, migrants who spent more than ten years in Cotonou are still having higher interbirth transition rate—although decreasing—as compared to non-migrants. This finding suggests that it is taking longer than expected for migrants to adjust to non-migrant's fertility behaviours. This adaptation process seems to be a long-run process. Therefore, this study supports the socialization hypothesis.

Even as this study offers valuable insights into how methodological strategies can fundamentally alter study conclusions, and offers support for the socialization hypotheses, it is important that we acknowledge the limitations of our results, including the extent to which they can be generalised to other settings in SSA remains uncertain. These results apply to Benin-born women, who may themselves have different fertility patterns, such as higher levels of fertility, as compared with women in other immigrant destinations in SSA. However, by demonstrating how fundamentally distinct conclusions are based on methodological approach, this study demonstrates the need to revisit this question across multiple contexts in sub-Saharan Africa, where questions of fertility determinants continue to be of prime interest.

## Acknowledgments

We acknowledge Emily Smith-Greenaway and Emmanuel Olamijuwon for reviewing and improving manuscript with their suggestions.

## Author Contributions

**Conceptualization:** Boladé Hamed Banougnin, Oluwaseyi Dolapo Somefun, Abibatou Agbéké Olakunle.

**Data curation:** Boladé Hamed Banougnin, Oluwaseyi Dolapo Somefun, Abibatou Agbéké Olakunle.

**Formal analysis:** Boladé Hamed Banougnin, Oluwaseyi Dolapo Somefun, Abibatou Agbéké Olakunle.

**Writing – original draft:** Boladé Hamed Banougnin, Oluwaseyi Dolapo Somefun, Abibatou Agbéké Olakunle.

**Writing – review & editing:** Boladé Hamed Banougnin, Oluwaseyi Dolapo Somefun, Abibatou Agbéké Olakunle.

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
