## [Decision Letter · Decision Letter 0]

20 Aug 2021

PONE-D-21-20332

Migration and Interbirth Transition Rate in Cotonou using Demographic and Health Survey Data: Does Episode-splitting Matter?

PLOS ONE

Dear Dr. Somefun,

Thank you for submitting your manuscript to PLOS ONE. After careful consideration, we feel that it has merit but does not fully meet PLOS ONE’s publication criteria as it currently stands. Therefore, we invite you to submit a revised version of the manuscript that addresses the points raised during the review process.

Two experts in the field handled your manuscript. We are appreciative of their time and contributions. Although interest was found in your study, some comments arose that require your attention. Some of these comments can be found in the attachment. 

We look forward to receiving your revised manuscript.

Kind regards,

Frank T. Spradley

Academic Editor

PLOS ONE

Reviewers' comments:

Reviewer's Responses to Questions

**Comments to the Author**

1. Is the manuscript technically sound, and do the data support the conclusions?

Reviewer #1: Yes

Reviewer #2: Yes

2. Has the statistical analysis been performed appropriately and rigorously? 

Reviewer #1: I Don't Know

Reviewer #2: Yes

3. Have the authors made all data underlying the findings in their manuscript fully available?

Reviewer #1: Yes

Reviewer #2: Yes

4. Is the manuscript presented in an intelligible fashion and written in standard English?

Reviewer #1: Yes

Reviewer #2: Yes

5. Review Comments to the Author

Reviewer #1: This is one of the most interesting papers I have had the pleasure to review in the field for some time. This is such an important topic, and it is hardly ever explored - not least because of methodological paucity. It is fantastic to see a genuine, systematic effort to link migration and fertility as INTERDEPENDENT factors.

I have to be completely honest and state that I am not familiar with the ins and outs of the exponential- regression splitting method deployed. In this sense, I defer to other reviewers. Subject to their approval, then, I would be very happy to see this in print.

I would only make a few small suggestions.

1. I think many readers may not be familiar with the situation in Benin. This could be explored in a bit more depth in the intro and conclusion.

2. The title refers explictly to Cotonou (not even Benin!) This may potentially put off readers who are not inherently interested in the city. I would suggest broadening the title to emphasise the method and the theme, with Benin as a case study or example. Maybe even just something like "Migration and Interbirth Transition Rates: New methodological insights" Basocially, try to sell it a bit more

3. Given the potential for this to be used elsewhere, I would really encourage the authors to supply their code for replication.

Reviewer #2: Please see the attachment.

6. PLOS authors have the option to publish the peer review history of their article (what does this mean?). If published, this will include your full peer review and any attached files.

Reviewer #1: No

Reviewer #2: No

---

## [Author Response · Author response to Decision Letter 0]

10 Sep 2021

Dear editor,

We are submitting a revised version of the manuscript “Migration and Interbirth Transition Rate in Cotonou using Demographic and Health Survey Data: Does Episode-splitting Matter?” (PONE-D-21-20332) for your appreciation and for editorial analysis by PLOS ONE, as to the possibility of its publication.

We thank the editor and reviewers for their comments and suggestions, which certainly have contributed to improve the paper. Detailed responses to each point raised by the reviewers are given below. We truly believe we have addressed the entire reviewers’ concerns.

Yours sincerely,

The authors

 

Reviewer #1: This is one of the most interesting papers I have had the pleasure to review in the field for some time. This is such an important topic, and it is hardly ever explored - not least because of methodological paucity. It is fantastic to see a genuine, systematic effort to link migration and fertility as INTERDEPENDENT factors.

I have to be completely honest and state that I am not familiar with the ins and outs of the exponential- regression splitting method deployed. In this sense, I defer to other reviewers. Subject to their approval, then, I would be very happy to see this in print.

I would only make a few small suggestions.

1. I think many readers may not be familiar with the situation in Benin. This could be explored in a bit more depth in the intro and conclusion.

Answer: Thank you for your suggestion. Changes are effected on Pages 6-7 with yellow marked-up

2. The title refers explicitly to Cotonou (not even Benin!) This may potentially put off readers who are not inherently interested in the city. I would suggest broadening the title to emphasise the method and the theme, with Benin as a case study or example. Maybe even just something like "Migration and Interbirth Transition Rates: New methodological insights" Basically, try to sell it a bit more

Answer: We appreciate this suggestion and modified the title. The new title is “Migration and interbirth transition rate using Benin Demographic and Health Survey Data: Does episode-splitting matter?” 

3. Given the potential for this to be used elsewhere, I would really encourage the authors to supply their code for replication.

Answer: Thank you for your suggestion. We will make code available along with the published paper

 

Reviewer #2:

Introduction

This article provides an interesting and detailed explanation of how episode splitting in migration studies affects estimates of inter-birth intervals, and the implications of this for our understanding of acclimatization of migrants to new settings. The methods are correct and the study conclusions justified from the results and methods. I have no significant concerns with the paper, and just a few comments which I hope the authors will address.

Comments

Comparison with standard time-varying covariates: It is not clear to me that the method employed here is different to a standard survival analysis with time-varying covariates, usually implemented in Stata using the stsplit command. Is the issue here that by using a directly-defined likelihood it is possible to adjust for survey sampling effects? Is the specific definition of an exponential distribution for the likelihood somehow unusual? I think the authors need to specify in more detail what new material their method adds. IN particular, if they have a specifically defined likelihood, how did they incorporate the survey sample process into that Likelihood? This is not specified in the methods. I think they just ran stsplit and then conducted survival analysis with a time-varying covariate? A little more detail about this in the methods would help to clarify the novelty of the study.

Answer: Thank you so much for this comment. Throughout the document, we used appropriate words making it clear that the episode-splitting approach is not new in the framework of survival analysis, but is almost not used the literature of migration-fertility relationship. Marked-up in yellow on Pages 11-12

Adjustment for wealth: I assume that the Benin DHS includes a variable for wealth quintile, and I wonder whether the effect of migration persists after adjusting for wealth quintile. Does wealth quintile change after migration? Were the migrant women more likely to be of a lower or high wealth quintile, with different birth intervals? The authors have adjusted for education but not for wealth quintile, which is standard practice in this kind of study. I think they should justify this decision or add this covariate to their model (sorry in advance if it makes the results non-significant; I still think the paper is publishable if a non-significant result arises, though obviously the discussion would need to change a lot).

Answer: Thank you for your suggestion. We adjusted for wealth quintile and updated Tables 3 and 4 on Pages 18 and 20. Changes in interpretation are also marked-up in yellow. 

Importance of the effect: In the basic model (not incorporating migration timing) the hazard ratios are quite small. What is the public health and demographic significance of these small hazard ratios? How much to they affect maternal health and welfare?

Answer: Thank you for raising this point. We believe this is not part of the aim of this study. The authors will think of investigating this aspect of the study in other occasion

---

## [Decision Letter · Decision Letter 1]

24 Sep 2021

Migration and interbirth transition rate using Benin Demographic and Health Survey data: Does episode-splitting matter?

PONE-D-21-20332R1

Dear Dr. Somefun,

We’re pleased to inform you that your manuscript has been judged scientifically suitable for publication and will be formally accepted for publication once it meets all outstanding technical requirements.

Kind regards,

Frank T. Spradley

Academic Editor

PLOS ONE

Reviewers' comments:

Reviewer's Responses to Questions

**Comments to the Author**

1. If the authors have adequately addressed your comments raised in a previous round of review and you feel that this manuscript is now acceptable for publication, you may indicate that here to bypass the “Comments to the Author” section, enter your conflict of interest statement in the “Confidential to Editor” section, and submit your "Accept" recommendation.

Reviewer #1: All comments have been addressed

2. Is the manuscript technically sound, and do the data support the conclusions?

Reviewer #1: Yes

3. Has the statistical analysis been performed appropriately and rigorously? 

Reviewer #1: Yes

4. Have the authors made all data underlying the findings in their manuscript fully available?

Reviewer #1: Yes

5. Is the manuscript presented in an intelligible fashion and written in standard English?

Reviewer #1: Yes

6. Review Comments to the Author

Reviewer #1: Love it. Can't wait to see it in print :) Happy with all responses to comments.

My only final suggestion - I think you can go even bigger on the title to get more people engaged. It still looks quite narrow. How about something as big as "The relationship between migration and fertility in Sub-Saharan Africa"

7. PLOS authors have the option to publish the peer review history of their article (what does this mean?). If published, this will include your full peer review and any attached files.

Reviewer #1: No

---

## [Editor Report · Acceptance letter]

13 Oct 2021

PONE-D-21-20332R1 

Migration and interbirth transition rate using Benin Demographic and Health Survey data: Does episode-splitting matter? 

Dear Dr. Somefun:

I'm pleased to inform you that your manuscript has been deemed suitable for publication in PLOS ONE. Congratulations! Your manuscript is now with our production department. 

Kind regards, 

on behalf of

Dr. Frank T. Spradley 

Academic Editor

PLOS ONE